# The Onsager–Machlup functional for data assimilation

Nozomi Sugiura[1]

[1]Research and Development Center for Global Change, JAMSTEC, Yokosuka, Japan

*Correspondence to:* Nozomi Sugiura (nsugiura@jamstec.go.jp)

**Abstract.** When taking the model error into account in data assimilation, one needs to evaluate the prior distribution represented by the Onsager–Machlup functional. Through numerical experiments, this study clarifies how the prior distribution should be incorporated into cost functions for discrete-time estimation problems. Consistent with previous theoretical studies, the divergence of the drift term is essential in weak-constraint 4D-Var (w4D-Var), but it is not necessary in Markov-chain Monte Carlo with the Euler scheme. Although the former property may cause difficulties when implementing w4D-Var in large systems, this paper proposes a new technique for estimating the divergence term and its derivative.

## 1  Introduction

In traditional weak-constraint 4D-Var settings (e.g. Zupanski, 1997; Trémolet, 2006), a quadratic cost function is defined as the negative logarithm of the probability for each sample path, which is suitable for path sampling (e.g. Zinn-Justin, 2002). The optimisation problem is naively described as finding the most probable path by minimising the quadratic cost function. However, the term 'the most probable path' does not make sense in this context, because the paths are not countable. One should notice that the concern is not about ranking the individual path probabilities, but about seeking the route with the densest path population. To define the optimisation problem properly, one should introduce a macroscopic variable $\phi = \phi(t)$ that represents a smooth curve, and introduce a measure that accounts for how densely the paths are populated in the $\epsilon$-neighbourhood centred at $\phi$, which can be termed as 'the tube'. Then the problem is defined as finding the most probable tube $\phi$, which represents the maximum a posteriori (MAP) estimate of the path distribution. Mathematicians pioneering the theory of stochastic differential equations (SDEs) (e.g. Ikeda and Watanabe, 1981; Zeitouni, 1989) have been aware of this subtle point since the 1980s, and established the proper form of the cost function as the Onsager–Machlup (OM) functional (Onsager and Machlup, 1953) for the tube.

The aim of this work is to organise existing knowledge about the OM functional into a form that can be used to represent model errors in data assimilation, i.e. numerical evaluation of nonlinear smoothing problems.

Throughout this article, we consider nonlinear smoothing problems of the form

$$dx_t = f(x_t)dt + \sigma dw_t, \tag{1}$$

$$x_0 \sim \mathcal{N}(x_b, \sigma_b^2 I), \tag{2}$$

$$(\forall m \in M) \quad y_m | x_m \sim \mathcal{N}(x_m, \sigma_o^2 I), \tag{3}$$

where $t$ is time, $x$ is a $D$-dimensional stochastic process, $w$ is a $D$-dimensional Wiener process, $x_b \in \mathbb{R}^D$ is the background value of the initial condition, $\sigma_b > 0$ is the standard deviation of the background value, $y_m \in \mathbb{R}^D$ is observational data at time $t_m$, $x_m = x_{t_m}$, $t_m = m\delta_t$, $M$ is the set of observation times, $\sigma_o > 0$ is the standard deviation of the observational data, and $\sigma > 0$ is the noise intensity. Note that there is no need to distinguish the Ito integral from the Stratonovich integral with regard to the discretisation of the SDE, because the noise intensity is a constant.

Before moving on to its applications, here we review the concept of the OM functional. To make presentation simple, we assume that $D = 1$ and $\sigma = 1$, and concentrate on the formulation of the prior distribution in the subsequent two sections 1.1 and 1.2.

## 1.1 OM functional for path sampling

The model equation (1) is discretised with the Euler scheme (with the drift term at the previous time) as

$$x_n = x_{n-1} + f(x_{n-1})\delta_t + \xi_{n-1}, \quad n = 1, 2, \cdots, N, \tag{4}$$

where $\delta_t$ is the time increment, and each $\xi_{n-1}$ obeys $\mathcal{N}(0, \delta_t)$. Equation (4) can be considered as a nonlinear mapping $F_1 : \xi \mapsto x$, from the noise vector $\xi = (\xi_0, \xi_1, \cdots, \xi_{N-1})^T$ to the state vector $x = (x_1, x_2, \cdots, x_N)^T$. The inverse of the mapping is linearised as

$$
\begin{bmatrix} \delta\xi_0 \\ \delta\xi_1 \\ \vdots \\ \delta\xi_{N-1} \end{bmatrix} =
\begin{bmatrix}
1 & 0 & \cdots & 0 & 0 \\
-1 - \delta_t f'(x_1) & 1 & & 0 & 0 \\
\vdots & & & & \vdots \\
0 & 0 & \cdots & -1 - \delta_t f'(x_{N-1}) & 1
\end{bmatrix}
\begin{bmatrix} \delta x_1 \\ \delta x_2 \\ \vdots \\ \delta x_N \end{bmatrix}, \tag{5}
$$

where $f'$ is the derivative of $f$, and the Jacobian is $DF_1^{-1} = |d\xi/dx| = 1$.

It is also discretised with the trapezoidal scheme (with the drift term at the midpoint) as

$$x_n = x_{n-1} + \frac{f(x_n) + f(x_{n-1})}{2}\delta_t + \xi_{n-1}, \quad n = 1, 2, \cdots, N, \tag{6}$$

which defines a mapping $F_2 : \xi \mapsto x$. The inverse of the mapping is linearised as

$$
\begin{bmatrix} \delta\xi_0 \\ \delta\xi_1 \\ \vdots \\ \delta\xi_{N-1} \end{bmatrix} =
\begin{bmatrix}
1 - \frac{\delta_t}{2} f'(x_1) & 0 & \cdots & 0 & 0 \\
-1 - \frac{\delta_t}{2} f'(x_1) & 1 - \frac{\delta_t}{2} f'(x_2) & & 0 & 0 \\
\vdots & & & & \vdots \\
0 & 0 & \cdots & -1 - \frac{\delta_t}{2} f'(x_{N-1}) & 1 - \frac{\delta_t}{2} f'(x_N)
\end{bmatrix}
\begin{bmatrix} \delta x_1 \\ \delta x_2 \\ \vdots \\ \delta x_N \end{bmatrix}, \tag{7}
$$

whose Jacobian is $DF_2^{-1} = |d\xi/dx| = \prod_{n=1}^{N} [1 - (\delta_t/2) f'(x_n)] \risingdotseq \exp\left[-(\delta_t/2) \sum_{n=1}^{N} f'(x_n)\right]$.

Generally, we can assign a measure $\mu_0$ to a cylinder set $\hat{\Omega} \equiv \hat{\Omega}_0 \times \hat{\Omega}_1 \times \cdots \times \hat{\Omega}_{N-1}$ in the noise space using a density $g$ as follows.

$$\mu_0(\hat{\Omega}) = \int_{\hat{\Omega}_0} d\xi_0 \int_{\hat{\Omega}_1} d\xi_1 \cdots \int_{\hat{\Omega}_{N-1}} d\xi_{N-1} g(\xi_0, \xi_1, \cdots, \xi_{N-1}) = \int_{\hat{\Omega}} g(\xi)\lambda(d\xi) = \int_{\hat{\Omega}} \mu_0(d\xi), \tag{8}$$

where $\lambda$ is the Lebesgue measure on $\mathbb{R}^N$. In our case, we can regard that a small area $d\xi$ in the noise space is equipped with a measure:

$$\mu_0(d\xi) = g(\xi)\lambda(d\xi), \quad g(\xi) \equiv \frac{1}{(2\pi\delta_t)^{N/2}}e^{-\frac{1}{2\delta_t}\sum_{n=1}^{N}\xi_{n-1}^2}. \tag{9}$$

Suppose we have a cylinder set $\Omega \equiv \Omega_1 \times \Omega_2 \times \cdots \times \Omega_N$ in the state space, where each $\Omega_n \subset \mathbb{R}^1$ is on time slice $t = n\delta_t$. Now, the mapping $F_1$ (or $F_2$) induces a measure through the change-of-variables from $\xi$ to $x$ with respect to the measure $\mu_0$ as

$$\mu_i(\Omega) = \int_{\Omega_1} dx_1 \int_{\Omega_2} dx_2 \cdots \int_{\Omega_N} dx_N (g \circ F_i^{-1})(x_1, x_2, \cdots, x_N)DF_i^{-1} = \int_\Omega \mu_i(dx), \quad i = 1, 2. \tag{10}$$

In our case, each mapping assigns the following measure to a small area $dx$ in the corresponding state space:

$$\mu_1(dx) \equiv g(F_1^{-1}(x))DF_1^{-1}\lambda(dx) = \frac{1}{(2\pi\delta_t)^{N/2}}e^{-\frac{\delta_t}{2}\sum_{n=1}^{N}\left(\frac{x_n-x_{n-1}}{\delta_t}-f(x_{n-1})\right)^2}\lambda(dx), \tag{11}$$

$$\mu_2(dx) \equiv g(F_2^{-1}(x))DF_2^{-1}\lambda(dx) = \frac{1}{(2\pi\delta_t)^{N/2}}e^{-\frac{\delta_t}{2}\sum_{n=1}^{N}\left[\left(\frac{x_n-x_{n-1}}{\delta_t}-f(x_{n-\frac{1}{2}})\right)^2+f'(x_n)\right]}\lambda(dx), \tag{12}$$

where $f(x_{n-\frac{1}{2}}) = \frac{f(x_n)+f(x_{n-1})}{2}$.

Measures $\mu_1$ and $\mu_2$ represent the occurrence probability of the noise seen from the state space, and thus can be used for path sampling.

The change-of-measure argument (Appendix B1) or the path integral argument (e.g. Zinn-Justin, 2002) shows that similar forms are available for time-continuous and multi-dimensional processes, except the term $f'(x_t)$ is promoted to $\operatorname{div} f(x_t)$.

## 1.2   OM functional for mode estimate

If we perform path sampling with a sufficient number of paths, in theory we can find the mean of distribution via averaging the samples, or the mode of distribution via organising them into a histogram. Still, in some practical applications, we must efficiently find the mode of distribution via variational methods; computationally, this approach is much cheaper than path sampling. For that purpose, we are tempted to use a quadratic cost function for the minimisation. However, we can illustrate a simple example against maximising the path probability (11) to obtain the mode of distribution. Suppose we have a discrete-time stochastic system in $\mathbb{R}^1$, starting from $x_0 = 0$, and we move forward two time steps:

$$x_1 = x_0 + x_0^2\delta_t + \xi_0 = \xi_0, \qquad x_2 = x_1 + x_1^2\delta_t + \xi_1 = \xi_0 + \xi_0^2\delta_t + \xi_1, \tag{13}$$

where $\xi_0$ and $\xi_1$ obey independent normal distributions $\mathcal{N}(0, \delta_t)$. It may be seen as a discrete version of $dx_t = x_t^2 dt + dw_t$. It is easy to notice that the mode of distribution $(x_1, x_2)$ is not $(0, 0)$ owing to the nonlinear term $\xi_0^2\delta_t$. On the other hand, according to the path probability (11):

$$\mu_1(dx_1 dx_2) \propto \exp\left[-\frac{\delta_t}{2}\left(\left(\frac{x_1-x_0}{\delta_t}-x_0^2\right)^2+\left(\frac{x_2-x_1}{\delta_t}-x_1^2\right)^2\right)\right]\lambda(dx_1 dx_2),$$

the best trajectory is $(x_1, x_2) = (0,0)$, which has no noise $(\xi_0, \xi_1) = (0,0)$. We expect a path with the highest probability at $(x_1, x_2) = (0,0)$, but it is not the route where the paths are most concentrated.

Motivated by this example, we shall investigate a proper strategy to find the route that maximises the density of paths. In this regard, we ask how densely the paths populate in the small neighbourhood of a curve $\phi = \phi(t)$ in the state space.

5      Assuming that $f$ and $\phi$ are twice continuously differentiable, we evaluate the density of paths in the $\epsilon$-neighbourhoods around a curve $\phi$ connecting points $\{\phi_n, \; n = 1, 2, \cdots, N\}$ with the following integral:

$$I_{\epsilon,\delta_t}(\phi) = \int_{\phi_1-\epsilon}^{\phi_1+\epsilon} dx_1 \int_{\phi_2-\epsilon}^{\phi_2+\epsilon} dx_2 \cdots \int_{\phi_N-\epsilon}^{\phi_N+\epsilon} dx_N \frac{1}{(2\pi\delta_t)^{N/2}} \exp\left\{ -\frac{\delta_t}{2} \sum_{n=1}^{N} \left( \frac{x_n - x_{n-1}}{\delta_t} - f(x_{n-1}) \right)^2 \right\} \tag{14}$$

$$= \int_{-\epsilon}^{\epsilon} dv_1 \int_{-\epsilon}^{\epsilon} dv_2 \cdots \int_{-\epsilon}^{\epsilon} dv_N \frac{1}{(2\pi\delta_t)^{N/2}} \exp\left\{ -\frac{\delta_t}{2} \sum_{n=1}^{N} \left( \frac{v_n - v_{n-1}}{\delta_t} + \frac{\phi_n - \phi_{n-1}}{\delta_t} - f(v_{n-1} + \phi_{n-1}) \right)^2 \right\} \tag{15}$$

$$= \int_{-\epsilon}^{\epsilon} dv_1 \int_{-\epsilon}^{\epsilon} dv_2 \cdots \int_{-\epsilon}^{\epsilon} dv_N \frac{1}{(2\pi\delta_t)^{N/2}} \exp\left\{ -\frac{\delta_t}{2} \sum_{n=1}^{N} \left( \frac{v_n - v_{n-1}}{\delta_t} \right)^2 \right\}$$

$$\times \exp\left\{ -\frac{\delta_t}{2} \sum_{n=1}^{N} \left[ \left( \frac{\phi_n - \phi_{n-1}}{\delta_t} - f(v_{n-1} + \phi_{n-1}) \right)^2 + 2 \left( \frac{\phi_n - \phi_{n-1}}{\delta_t} - f(v_{n-1} + \phi_{n-1}) \right) \left( \frac{v_n - v_{n-1}}{\delta_t} \right) \right] \right\}. \tag{16}$$

By regarding $v_n$ in Eq. (16) as being generated according to the probability $\frac{1}{(2\pi\delta_t)^{N/2}} e^{-\frac{\delta_t}{2} \sum_{n=1}^{N} \left( \frac{v_n - v_{n-1}}{\delta_t} \right)^2}$, we can interpret the integration as a weighted ensemble averaging of a random function up to a numerical constant. The sequence $v_n$ can be set as a random walk $v_0 = 0$, $v_n = \sum_{k=1}^{n} \xi_k$, where $\xi_k$ are independent normal random variables obeying $\mathcal{N}(0, \delta_t)$. For simplicity, we rather assume that $\xi_k$ takes values $\pm\sqrt{\delta_t}$ with 0.5 probability for either one, because Donsker's theorem ensures it has

15 the same probability law as the former when $\delta_t$ is sufficiently small. We suppose $\sqrt{\delta_t} < \epsilon$ so that no step of the random walk escapes from the $\epsilon$-neighbourhood. Accordingly, the integral is expressed as the ensemble average with respect to random walks confined in the tube $[0, N\delta_t] \times [-\epsilon, \epsilon]$:

$$I_{\epsilon,\delta_t}(\phi) \propto \mathbb{E}_v \left[ e^{-J(\phi,v)} \big| (\forall n) \; |v_n| < \epsilon \right], \tag{17}$$

$$J(\phi, v) \equiv \frac{\delta_t}{2} \sum_{n=1}^{N} \left[ \left( \frac{\phi_n - \phi_{n-1}}{\delta_t} - f(v_{n-1} + \phi_{n-1}) \right)^2 + 2 \left( \frac{\phi_n - \phi_{n-1}}{\delta_t} - f(v_{n-1} + \phi_{n-1}) \right) \left( \frac{v_n - v_{n-1}}{\delta_t} \right) \right] \tag{18}$$

20 where $\mathbb{E}_v$ denotes the ensemble averaging of the random walks denoted by $v$, each of which follows the route $(v_0, v_1, \cdots, v_N)$, and satisfies $|v_n| < \epsilon$ for all $n$.

Because $v_{n-1}$ is small, we can apply the expansion:

$$f(v_{n-1} + \phi_{n-1}) = f(\phi_{n-1}) + f'(\phi_{n-1})v_{n-1} + O(v^2), \tag{19}$$

where $f'$ is the derivative of $f$. Let us accept that the following average containing the higher order terms $O(v^2)$ converges (see Eq. (B20)).

$$\mathbb{E}_v\left[e^{\sum_{n=1}^N O(v^2)(v_n - v_{n-1})}\Big| (\forall n) \ |v_n| < \epsilon\right] \xrightarrow{\epsilon \to 0} 1. \tag{20}$$

As shown in Appendix B2, the remaining terms in the exponent $-J(\phi, v)$ are less than $O(\epsilon)$ except the following one.

$$5 \quad \sum_{n=1}^N f'(\phi_{n-1})v_{n-1}(v_n - v_{n-1}) = \sum_{n=1}^N f'(\phi_{n-1})\left[\frac{1}{2}(v_{n-1} - v_n) + \frac{1}{2}(v_{n-1} + v_n)\right](v_n - v_{n-1}) \tag{21}$$

$$= \sum_{n=1}^N f'(\phi_{n-1})\frac{1}{2}(v_{n-1} - v_n)(v_n - v_{n-1}) + \sum_{n=1}^N f'(\phi_{n-1})\frac{1}{2}(v_n^2 - v_{n-1}^2) \tag{22}$$

$$= -\frac{1}{2}\sum_{n=1}^N f'(\phi_{n-1})\xi_n^2 + \frac{1}{2}\sum_{n=1}^{N-1}[f'(\phi(t_{n-1})) - f'(\phi(t_{n-1} + \delta_t))]v_n^2 + \frac{1}{2}f'(\phi_{N-1})v_N^2 \tag{23}$$

$$= -\frac{\delta_t}{2}\sum_{n=1}^N f'(\phi_{n-1}) + O(\epsilon^2). \qquad \because \xi_n = \pm\sqrt{\delta_t}, \quad f'(\phi(t_{n-1})) - f'(\phi(t_{n-1} + \delta_t)) = O(\delta_t), \quad v_n^2 < \epsilon^2. \tag{24}$$

Consequently, we obtain the asymptotic expression for the ensemble average when $\epsilon$ is small and $\delta_t < \epsilon^2$:

$$10 \quad I_{\epsilon, \delta_t}(\phi) \propto \mathbb{E}_v\left[e^{-\frac{\delta_t}{2}\sum_{n=1}^N\left[\left(\frac{\phi_n - \phi_{n-1}}{\delta_t} - f(\phi_{n-1})\right)^2 + f'(\phi_{n-1})\right] + O(\epsilon) + \sum_{n=1}^N O(v^2)(v_n - v_{n-1})}\Big| (\forall n) \ |v_n| < \epsilon\right] \tag{25}$$

$$\to e^{-\frac{1}{2}\int_0^T\left[\left(\dot\phi(t) - f(\phi(t))\right)^2 + f'(\phi(t))\right]dt}. \tag{26}$$

Appendix B2 shows that a similar form is available for time-continuous and multi-dimensional processes, except the term $f'(\phi(t))$ is promoted to $\operatorname{div} f(\phi(t))$.

Importantly, the control variable for the optimisation has changed from $x$ to $\phi$.

## 15   1.3   Probabilistic description of data assimilation

Using the OM functional derived in sections 1.1 and 1.2 as a model error term, we shall develop a probabilistic description of data assimilation.

Following the derivation in section 2.3 of Law et al. (2015), we can assign each path a posterior probability

$$P(x|y) \propto P(x)P(y|x) = P(x|x_0)P(x_0)P(y|x) = \prod_{n=1}^N P(x_n|x_{n-1})P(x_0)\prod_{m \in M} P(y_m|x_m). \tag{27}$$

20   According to Eq. (2), the prior probability for the initial condition is given as

$$P(x_0) \propto \exp\left(-\frac{|x_0 - x_b|^2}{2\sigma_b^2}\right), \tag{28}$$

where $|x_0 - x_b|^2$ represents the squared Euclidean norm $\sum_{i=1}^D (x_0^i - x_b^i)^2$. According to Eq. (3), the likelihood of the state $x_m$, given observation $y_m$, is

$$P(y_m|x_m) \propto \exp\left(-\frac{|y_m - x_m|^2}{2\sigma_o^2}\right). \tag{29}$$

Based on the argument in section 1.1, Eq. (4) has the transition probability at discrete time steps

$$P(x_n|x_{n-1}) \propto \exp\left(-\frac{\delta_t}{2\sigma^2}\left|\frac{x_n - x_{n-1}}{\delta_t} - f(x_{n-1})\right|^2\right), \tag{30}$$

called the Euler scheme, which uses the drift $f(x_{n-1})$ at the previous time step. Section 1.1 also shows that this transition probability has another expression:

$$P(x_n|x_{n-1}) \propto \exp\left(-\frac{\delta_t}{2\sigma^2}\left|\frac{x_n - x_{n-1}}{\delta_t} - f(x_{n-\frac{1}{2}})\right|^2 - \frac{\delta_t}{2}\operatorname{div} f(x_n)\right), \tag{31}$$

$$f(x_{n-\frac{1}{2}}) \equiv \frac{f(x_n) + f(x_{n-1})}{2}, \quad \operatorname{div} f(x) \equiv \sum_{i=1}^{D}\frac{\partial f^i}{\partial x^i}(x), \tag{32}$$

which can be called the trapezoidal scheme because the integral is evaluated with the drift terms at both ends of each interval. The transition probability leads to the prior probability $P(x|x_0)$ of a path $x = \{x_n\}_{0 \le n \le N}$ as follows:

$$P(x|x_0) \propto \exp\left(-\delta_t \sum_{n=1}^{N}\frac{1}{2\sigma^2}\left|\frac{x_n - x_{n-1}}{\delta_t} - f(x_{n-1})\right|^2\right) \tag{33}$$

$$\rightleftharpoons \exp\left(-\delta_t \sum_{n=1}^{N}\left[\frac{1}{2\sigma^2}\left|\frac{x_n - x_{n-1}}{\delta_t} - f(x_{n-\frac{1}{2}})\right|^2 + \frac{1}{2}\operatorname{div} f(x_n)\right]\right), \tag{34}$$

where '$\rightleftharpoons$' sign indicates that, if $\delta_t$ is sufficiently small, the equations on the both sides are compatible.

On the other hand, based on the argument in section 1.2, we can also define the probability $P(U_\phi|\phi_0)$ for a smooth tube that represents its neighbouring paths $U_\phi = \{\omega|(\forall n)|\phi_n - x_n(\omega)| < \epsilon\}$:

$$P(U_\phi|\phi_0) \propto \exp\left(-\delta_t \sum_{n=1}^{N}\left[\frac{1}{2\sigma^2}\left|\frac{\phi_n - \phi_{n-1}}{\delta_t} - f(\phi_{n-1})\right|^2 + \frac{1}{2}\operatorname{div} f(\phi_{n-1})\right]\right). \tag{35}$$

The scaling argument for a smooth curve in Appendix A allows us to use the drift term $f(\phi_{n-\frac{1}{2}})$ instead in Eq. (35):

$$P(U_\phi|\phi_0) \propto \exp\left(-\delta_t \sum_{n=1}^{N}\left[\frac{1}{2\sigma^2}\left|\frac{\phi_n - \phi_{n-1}}{\delta_t} - f(\phi_{n-\frac{1}{2}})\right|^2 + \frac{1}{2}\operatorname{div} f(\phi_{n-\frac{1}{2}})\right]\right). \tag{36}$$

The corresponding posterior probabilities are thus given as follows:

$$P_{\text{path}}(x|y) \propto \exp\left(-J_{\text{path}}(x|y)\right), \tag{37}$$

$$J_{\text{path}}(x|y) \equiv \frac{1}{2\sigma_b^2}|x_0 - x_b|^2 + \sum_{m \in M}\frac{1}{2\sigma_o^2}|x_m - y_m|^2 + \delta_t \sum_{n=1}^{N}\left(\frac{1}{2\sigma^2}\left|\frac{x_n - x_{n-1}}{\delta_t} - f(x_{n-1})\right|^2\right) \tag{38}$$

$$\rightleftharpoons \frac{1}{2\sigma_b^2}|x_0 - x_b|^2 + \sum_{m \in M}\frac{1}{2\sigma_o^2}|x_m - y_m|^2 + \delta_t \sum_{n=1}^{N}\left(\frac{1}{2\sigma^2}\left|\frac{x_n - x_{n-1}}{\delta_t} - f(x_{n-\frac{1}{2}})\right|^2 + \frac{1}{2}\operatorname{div} f(x_n)\right) \tag{39}$$

for a sample path, and

$$P_{\text{tube}}(U_\phi|y) \propto P(U_\phi|\phi_0)P(\phi_0)P(y|U_\phi) \propto \exp\left(-J_{\text{tube}}(\phi|y)\right), \tag{40}$$

$$J_{\text{tube}}(\phi|y) \equiv \frac{1}{2\sigma_b^2}|\phi_0 - x_b|^2 + \sum_{m \in M}\frac{1}{2\sigma_o^2}|\phi_m - y_m|^2 + \delta_t \sum_{n=1}^{N}\left(\frac{1}{2\sigma^2}\left|\frac{\phi_n - \phi_{n-1}}{\delta_t} - f(\phi_{n-\frac{1}{2}})\right|^2 + \frac{1}{2}\text{div}\, f(\phi_{n-\frac{1}{2}})\right) \tag{41}$$

$$\Leftarrow \frac{1}{2\sigma_b^2}|\phi_0 - x_b|^2 + \sum_{m \in M}\frac{1}{2\sigma_o^2}|\phi_m - y_m|^2 + \delta_t \sum_{n=1}^{N}\left(\frac{1}{2\sigma^2}\left|\frac{\phi_n - \phi_{n-1}}{\delta_t} - f(\phi_{n-1})\right|^2 + \frac{1}{2}\text{div}\, f(\phi_{n-1})\right) \tag{42}$$

for a smooth tube. Note that different pairs of time-discretisation schemes of the OM functional, $\frac{1}{2\sigma^2}\left(\frac{dx}{dt} - f(x)\right)^2 + \frac{1}{2}\text{div}\,(f)$, are nominated for paths and for tubes in Eqs. (38), (39), (41), and (42).

## 2  Method

### 2.1  Four schemes for OM

In the argument in sections 1.1 and 1.2, the prior probability has a form $P(x|x_0) \propto \exp\left(-\delta_t \sum_{n=1}^{N}\widetilde{OM}\right)$, where $\widetilde{OM}$ is the OM functional. As a proof-of-concept described in these sections, we will test all the cases with conceivable combinations of the timing of the drift term $f(x_t)$ and the presence or absence of the divergence term. Including those shown in Eqs. (38), (39), (41), and (42), as well as those that are potentially incorrect, the possible candidates for the discretisation schemes of the OM functional are as follows, where the symbol $\psi$ represents either $\phi$ for a smooth curve or $x$ for a sample path.

1. Euler scheme (E) (e.g. Zinn-Justin, 2002; Dutra et al., 2014):

$$\widetilde{OM}_{\text{E}} \equiv \frac{1}{2\sigma^2}\left|\frac{\psi_n - \psi_{n-1}}{\delta_t} - f(\psi_{n-1})\right|^2; \tag{43}$$

2. Euler scheme with divergence term (ED):

$$\widetilde{OM}_{\text{ED}} \equiv \frac{1}{2\sigma^2}\left|\frac{\psi_n - \psi_{n-1}}{\delta_t} - f(\psi_{n-1})\right|^2 + \frac{1}{2}\text{div}\, f(\psi_{n-1}); \tag{44}$$

3. Trapezoidal scheme (T):

$$\widetilde{OM}_{\text{T}} \equiv \frac{1}{2\sigma^2}\left|\frac{\psi_n - \psi_{n-1}}{\delta_t} - f(\psi_{n-\frac{1}{2}})\right|^2; \tag{45}$$

4. Trapezoidal scheme with divergence term (TD) (e.g. Ikeda and Watanabe, 1981; Apte et al., 2007; Dutra et al., 2014):

$$\widetilde{OM}_{\text{TD}} \equiv \frac{1}{2\sigma^2}\left|\frac{\psi_n - \psi_{n-1}}{\delta_t} - f(\psi_{n-\frac{1}{2}})\right|^2 + \frac{1}{2}\text{div}\, f(\psi_{n-\frac{1}{2}}), \tag{46}$$

where $f(\psi_{n-\frac{1}{2}}) = (f(\psi_n) + f(\psi_{n-1}))/2$.

## 2.2 Data assimilation algorithms

By using one of the above schemes adopted for the model error term in the cost function, we can apply a data assimilation algorithm—either Markov-chain Monte Carlo (MCMC) (e.g. Metropolis et al., 1953) or four-dimensional variational data assimilation (4D-Var) (e.g. Zupanski, 1997). Among versions of MCMC, we focus on the Metropolis-adjusted Langevin algorithm (MALA) (e.g. Roberts and Rosenthal, 1998; Cotter et al., 2013). MALA samples the paths $x^{(k)} = \{x_n(\omega_k)\}_{0 \leq n \leq N}$ according to the distribution $P_{\text{path}}$ by iterating:

$$x^{(k+1)} = x^{(k)} - \alpha \nabla J_{\text{path}} + \sqrt{2\alpha}\xi, \quad \alpha > 0, \ \xi \sim \mathcal{N}(0,1)^{D(N+1)}, \ \nabla J = \left(\frac{\partial J}{\partial x}\right)^T \tag{47}$$

with the Metropolis rejection step for adjustment, to obtain an ensemble of sample paths according to the posterior probability, while 4D-Var seeks the centre of the most probable tube $\phi = \{\phi_n\}_{0 \leq n \leq N}$ by iterating:

$$\phi^{(k+1)} = \phi^{(k)} - \alpha \nabla J_{\text{tube}}, \quad \alpha > 0. \tag{48}$$

Note that if the OM functional of type $\widetilde{OM}_{\text{ED}}$ is used, the gradient is of the form:

$$\begin{aligned}
\nabla_{\phi_n} J_{\text{tube}} &= \frac{1}{\sigma_b^2}(\phi_0 - x_b)\delta_{0,n} + \sum_{m \in M}\frac{1}{\sigma_o^2}(\phi_m - y_m)\delta_{m,n} \\
&+ \frac{1}{\sigma^2}\left(\frac{\phi_n - \phi_{n-1}}{\delta_t} - f(\phi_{n-1})\right) \qquad (n > 0) \\
&+ \frac{\delta_t}{\sigma^2}\left(-\frac{1}{\delta_t} - \left(\frac{\partial f}{\partial \phi_n}(\phi_n)\right)^T\right)\left(\frac{\phi_{n+1} - \phi_n}{\delta_t} - f(\phi_n)\right) + \frac{\delta_t}{2}\frac{\partial}{\partial \phi_n}\operatorname{div} f(\phi_n) \qquad (n < N),
\end{aligned} \tag{49}$$

where $\left(\frac{\partial f}{\partial \phi_n}(\phi_n)\right)^T$ is an adjoint integration starting from the subsequent term, which is typical in gradient calculations in 4D-Var. In comparison, the term $\frac{\partial}{\partial \phi_n}\operatorname{div} f(\phi_n)$ requires the second derivative of $f$, which is not typical in 4D-Var, and could be difficult to implement in large dimensional systems.

To investigate the applicability of the four candidate schemes in section 2.1, we use them in these algorithms.

The results should be checked with 'the correct answer'. The reference solution that approximates the correct answer is provided by a particle smoother (PS) (e.g. Doucet et al., 2000), which does not involve the explicit computation of prior probability. When we have observations only at the end of the assimilation window, the PS algorithm is as follows:

1. Generate samples of initial and model errors, integrate $M$ copies of the model, and use them to obtain a Monte-Carlo approximation of the prior distribution:

$$P(x) \simeq \frac{1}{M}\sum_{m=1}^{M}\prod_{n=0}^{N}\delta(x_n - \chi_n^{(m)}), \tag{50}$$

where $\chi_n^{(m)}$ is the state of member $m$ at time $n$.

2. Reweight it according to Bayes' theorem:

$$P(y|x) \propto \exp\left(-\frac{1}{2\sigma_o^2}|y-x_N|^2\right), \tag{51}$$

$$P(x|y) = \frac{P(x)P(y|x)}{\int dx P(x)P(y|x)} = \sum_{m=1}^{M}\prod_{n=0}^{N}\delta(x_n - \chi_n^{(m)})\frac{w^{(m)}}{\sum_{m=1}^{M}w^{(m)}}, \tag{52}$$

$$w^{(m)} \equiv \exp\left(-\frac{1}{2\sigma_o^2}|y-\chi_N^{(m)}|^2\right). \tag{53}$$

## 3 Results

### 3.1 Example A (hyperbolic model)

In our first example, we solve the nonlinear smoothing problem for the hyperbolic model (Daum, 1986), which is a simple problem with one-dimensional state space, but which has a nonlinear drift term. We want to find the probability distribution of the paths described by

$$dx_t = \tanh(x_t)dt + dw_t, \quad x_{t=0} \sim \mathcal{N}(0, 0.16), \tag{54}$$

subject to an observation $y$:

$$y|x_{t=5} \sim \mathcal{N}(x_{t=5}, 0.16), \quad y = 1.5. \tag{55}$$

The setting follows Daum (1986). In this case, $\operatorname{div} f(x) = 1/\cosh^2(x)$ imposes a penalty for small $x$. The total time duration $T = 5$ is divided into $N = 100$ segments with $\delta_t = 5 \times 10^{-2}$.

Figure 1 shows the probability densities of paths normalised on each time slice, $P_{t=n}(\phi) = \int P(U_\phi|y)d\phi_{t\neq n}$, derived by MCMC and PS. PS is performed with $5.1 \times 10^{10}$ particles. It is clear that MCMC with E or TD provides the proper distribution matched with that of PS; this is also clear from the expected paths yielded by these experiments, as shown in Fig. 2. These schemes correspond to candidates in Eqs. (38) and (39). The expected path by ED bends towards a larger $x$, which should be caused by an extra penalty for a larger $x$. The expected path by T bends towards a smaller $x$, which should be caused by the lack of a penalty for a larger $x$.

The results of 4D-Var, which represents the MAP estimates, are shown in Fig. 3. ED and TD provide the proper MAP estimate. These schemes correspond to candidates in Eqs. (41) and (42). The expected paths by E and T bend towards a smaller $\phi$, which should be caused by the lack of a penalty for a larger $\phi$.

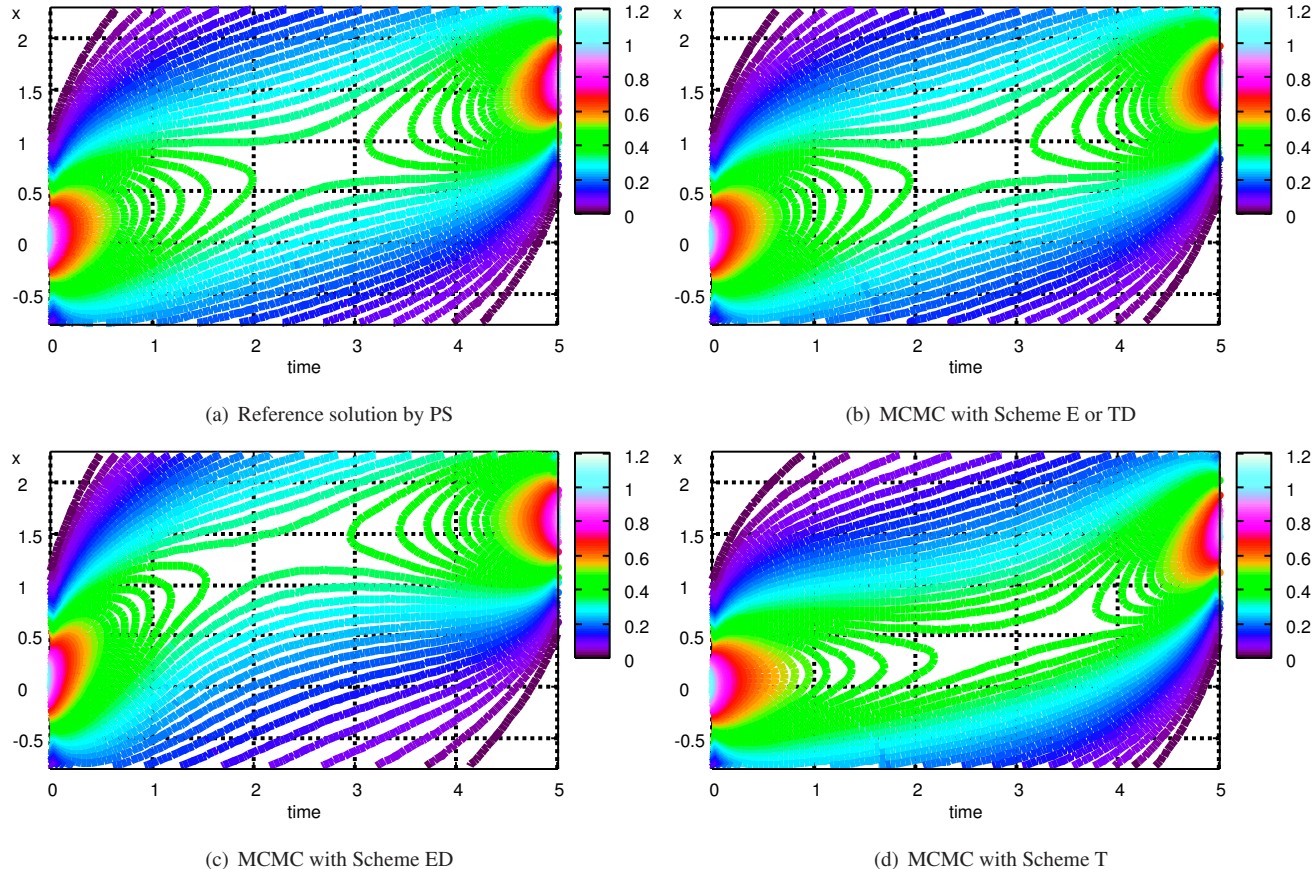

(a) Reference solution by PS

(b) MCMC with Scheme E or TD

(c) MCMC with Scheme ED

(d) MCMC with Scheme T

**Figure 1.** Probability density of paths derived by MCMC and PS for the hyperbolic model.

### 3.2 Example B (Rössler model)

In our second example, we solve the nonlinear smoothing problem for the stochastic Rössler model (Rössler, 1976). We want to find the probability distribution of the paths described by

$$
\begin{cases}
dx_1 &= (-x_2 - x_3)dt + \sigma dw_1, \\
dx_2 &= (x_1 + ax_2)dt + \sigma dw_2, \\
dx_3 &= (b + x_1 x_3 - cx_3)dt + \sigma dw_3,
\end{cases}
\tag{56}
$$

$$
x_{t=0} \sim \mathcal{N}(x_b, 0.04I),
\tag{57}
$$

subject to an observation $y$:

$$
y|x_{t=0.4} \sim \mathcal{N}(x_{t=0.4}, 0.04I),
\tag{58}
$$

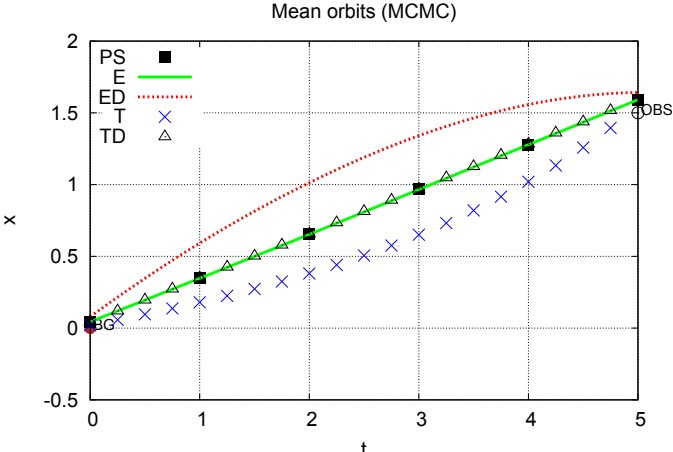

**Figure 2.** Expected path derived by MCMC (hyperbolic model).

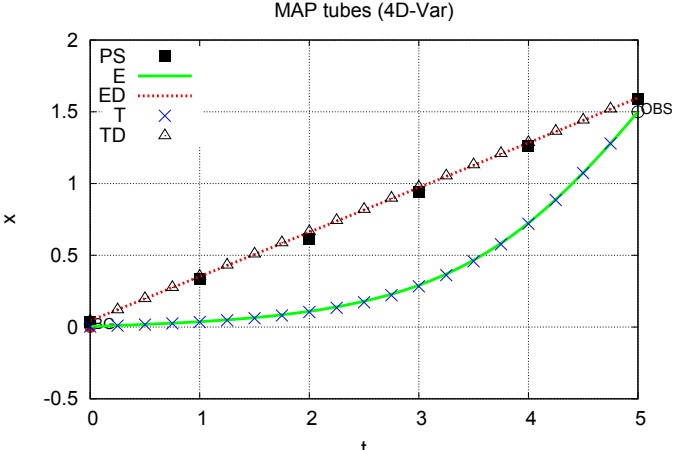

**Figure 3.** Most probable tube derived by 4D-Var (hyperbolic model).

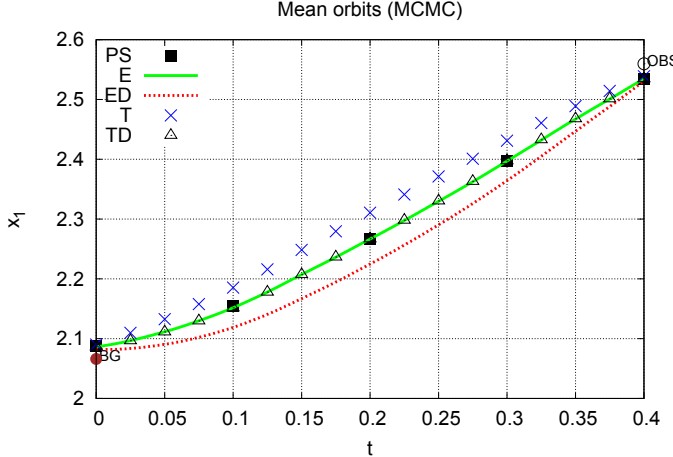

**Figure 4.** Expected path derived by MCMC (Rössler model).

where $(a, b, c) = (0.2, 0.2, 6)$, $\sigma = 2$, $x_b = (2.0659834, -0.2977757, 2.0526298)^T$, and $y = (2.5597086, 0.5412736, 0.6110939)^T$. In this case, $\operatorname{div} f(x) = x_1 + a - c$ imposes a penalty for large $x_1$. The total time duration $T = 0.4$ is divided into $N = 800$ segments with $\delta_t = 5 \times 10^{-4}$.

The results by MCMC and 4D-Var for the Rössler model are shown in Figs. 4 and 5, respectively. The state variable $x_1$ is chosen for the vertical axes. PS is performed with $3 \times 10^{12}$ particles. The curve for PS in Fig. 5 indicates $\hat{\phi} = \operatorname{argmax}_\phi P(\phi | y)$, where $U$ represents the tube centred at $\phi$ with radius $0.03$.

Figure 4 shows that, just as for the hyperbolic model, E and TD provide the proper expected path. Figure 5 shows that ED and TD provide the proper MAP estimate.

### 3.3 Towards application to large systems

When one computes the cost value $J(x)$, the negative logarithm of the posterior probability, in data assimilation, the value $f(x)$ is explicitly computed via the numerical model, while $\operatorname{div} f(x)$ is not. If the dimension $D$ of the state space is large, and $f$ is complicated, the algebraic expression of $\operatorname{div} f(x)$ can be difficult to obtain. The gradient of the cost function $\nabla J(x)$ contains the derivative of $f(x)$, which can be implemented as the adjoint model via symbolic differentiation (e.g. Giering and Kaminski, 1998). However, schemes with the divergence term require the calculation of the second derivative of $f(x)$, for which the algebraic expression can be even more difficult to obtain. Still, there may be a way to circumvent this difficulty by utilising Hutchinson's trace estimator (Hutchinson, 1990) (See Appendix C). It is also clear that the Euler scheme without the divergence term is more convenient for implementing path sampling, because it does not require cumbersome calculation of the divergence term.

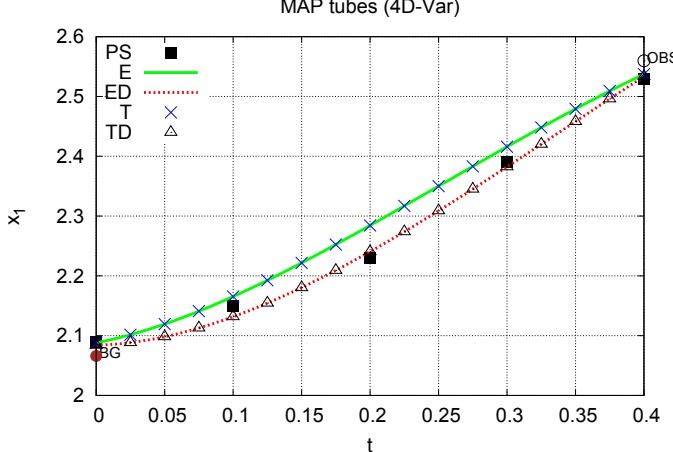

**Figure 5.** Most probable tube derived by 4D-Var (Rössler model).

## 4 Conclusions

We examined several discretisation schemes of the OM functional, $\frac{1}{2\sigma^2}\left(\frac{dx}{dt} - f(x)\right)^2 + \frac{1}{2}\mathrm{div}\,(f)$, for the nonlinear smoothing problem

$$dx_t = f(x_t)dt + \sigma dw_t,$$

$$x_0 \sim \mathcal{N}(x_b, \sigma_b^2 I), \quad (\forall m \in M)\, y_m | x_m \sim \mathcal{N}(x_m, \sigma_o^2 I)$$

by matching the answers given by MCMC and 4D-Var with that given by PS, taking the hyperbolic model and the Rössler model as examples. Table 1 lists the discretisation schemes which were found to be applicable, i.e. those expected to converge to the same result as the reference solution. These results are consistent with the literature (e.g. Apte et al., 2007; Malsom and Pinski, 2016; Dutra et al., 2014; Stuart et al., 2004).

This justifies, for instance, the use of the following cost function for the MAP estimate given by 4D-Var:

$$J = \frac{|\phi_0 - x_b|^2}{2\sigma_b^2} + \sum_{m \in M} \frac{|\phi_m - y_m|^2}{2\sigma_o^2}$$

$$+ \delta_t \sum_{n=1}^{N} \left( \frac{1}{2\sigma^2} \left| \frac{\phi_n - \phi_{n-1}}{\delta_t} - f(\phi_{n-1}) \right|^2 + \frac{1}{2}\mathrm{div}\,f(\phi_{n-1}) \right),$$

where $n$ is the time index, $\delta_t$ is the time increment, $x_b$ is the background value, $\sigma_b$ is the standard deviation of the background value, $y$ is the observational data, $\sigma_o$ is the standard deviation of the observational data, and $\sigma$ is the noise intensity. However,

the divergence term above should be excluded for the assignment of path probability in MCMC.

**Table 1.** Applicable OM schemes

|  |  | with div $(f)$ | without div $(f)$ |
|---|---|---|---|
| Sampling by MCMC | Euler scheme |  | ✓ |
|  | trapezoidal scheme | ✓ |  |
| MAP estimate by 4D-Var | Euler scheme | ✓ |  |
|  | trapezoidal scheme | ✓ |  |

For application in large systems, the Euler scheme without the divergence term is preferred for path sampling because it does not require cumbersome calculation of the divergence term. In 4D-Var, the divergence term can be incorporated into the cost function by utilising Hutchinson's trace estimator.

*Code availability.* The code for data assimilation is available at https://github.com/nozomi-sugiura/OnsagerMachlup/.

## 5   Appendix A: Scaling of the terms

Taylor expansion of the $f(\psi_{n-1})$ term around $\psi_{n-\frac{1}{2}}$ in scheme E gives

$$\widetilde{OM} \simeq \sum_{n=1}^{N} \delta_t \left\{ \sigma^{-2} \left[ \frac{\psi_n - \psi_{n-1}}{\delta_t} - f(\psi_{n-\frac{1}{2}}) - (\psi_n - \psi_{n-1})\frac{\partial f}{\partial x}(\psi_{n-\frac{1}{2}}) \right]^2 + \mathrm{div}\,(f) \right\}$$

$$= \delta_t \left\{ \sigma^{-2}(\mathrm{noise} + \mathrm{shift})^2 + \mathrm{divergence} \right\}.$$

$$\mathrm{noise} \equiv \frac{\psi_n - \psi_{n-1}}{\delta_t} - f(\psi_{n-\frac{1}{2}}), \mathrm{shift} \equiv (\psi_n - \psi_{n-1})\frac{\partial f}{\partial x}(\psi_{n-\frac{1}{2}}), \mathrm{divergence} \equiv \mathrm{div}\,(f),$$

10    where we assume order-one fluctuations: $\sigma = O(1)$, and the symbol $\psi$ represents either $\phi$ for a smooth curve or $x$ for a sample path.

For a sample path of the stochastic process, the scaling $\psi_n - \psi_{n-1} = O(\delta_t^{\frac{1}{2}})$, which leads to

$$\widetilde{OM} = \sum \delta_t \left\{ \sigma^{-2} \left( \underbrace{\mathrm{noise}^2}_{\delta_t^{-1}} + \underbrace{\mathrm{noise} \times \mathrm{shift}}_{1} + \underbrace{\mathrm{shift}^2}_{\delta_t} \right) + \underbrace{\mathrm{divergence}}_{1} \right\}. \tag{A1}$$

The shift term induces a Jacobian that coincides with the divergence term in TD (Zinn-Justin, 2002).

15    In the case of a smooth curve, there is no stochastic term, and thus $\psi_n - \psi_{n-1}$ is the product of a bounded function $f(\psi_{n-1})$ and $\delta_t$, which results in a value with $O(\delta_t)$. This leads to

$$\widetilde{OM} = \sum \delta_t \left\{ \sigma^{-2} \left( \underbrace{\mathrm{noise}^2}_{1} + \underbrace{\mathrm{noise} \times \mathrm{shift}}_{\delta_t} + \underbrace{\mathrm{shift}^2}_{\delta_t^2} \right) + \underbrace{\mathrm{divergence}}_{1} \right\}. \tag{A2}$$

The shift term is negligible, but the divergence term is not.

### Appendix B:  Divergence term

#### B1   Divergence term in a trapezoidal scheme

Consider two stochastic processes (cf., section 6.3.2 of Law et al. (2015)):

$$dx_t = f(x_t)dt + dw_t, \quad x(0) = x_0,$$  (B1)

$$dx_t = dw_t, \quad x(0) = x_0,$$  (B2)

where (B1) has measure $\mu$ and (B2) has measure $\mu_0$ (Wiener measure). By the Girsanov theorem, the Radon–Nikodym derivative of $\mu$ with respect to $\mu_0$ is

$$\frac{d\mu}{d\mu_0} = \exp\left[-\int_0^T \left(\frac{1}{2}|f(x)|^2 dt - f(x) \cdot dx\right)\right].$$  (B3)

If we define $F(x_T) - F(x_0) = \int_{x_0}^{x_T} f(x) \circ dx$ with the Stratonovich integral, then by Ito's formula,

$$dF = f \cdot dx + \frac{1}{2}\operatorname{div}(f)dt.$$  (B4)

Eliminating $f \cdot dx$ in Eq. (B3) using Eq. (B4), we obtain

$$\frac{d\mu}{d\mu_0} = \exp\left[-\int_0^T \frac{1}{2}|f(x)|^2 dt + F(x_T) - F(x_0) - \frac{1}{2}\int_0^T \operatorname{div}(f)dt\right].$$  (B5)

Substituting $F(x_T) - F(x_0) = \int_0^T f \circ \frac{dx}{dt}dt$,

$$\frac{d\mu}{d\mu_0} = \exp\left[-\int_0^T \frac{1}{2}|f(x)|^2 dt + \int_0^T f \circ \frac{dx}{dt}dt - \frac{1}{2}\int_0^T \operatorname{div}(f)dt\right].$$  (B6)

If we write the Wiener measure formally as $\mu_0(dx) = \exp\left[-\frac{1}{2}\int_0^T \left|\frac{dx}{dt}\right|^2 dt\right]dx$, we get the following from Eq. (B3)

$$\mu(dx) = \exp\left[-\int_0^T \frac{1}{2}\left|\frac{dx}{dt} - f(x)\right|^2 dt\right]dx$$  (B7)

and the following from Eq. (B6)

$$\mu(dx) = \exp\left[-\int_0^T \frac{1}{2}\left(\left|\frac{dx}{dt} - f(x)\right|^2 + \operatorname{div}(f)\right)dt\right]dx,$$  (B8)

where the integrals should be interpreted in the Ito sense and in the Stratonovich sense, respectively.

## B2 Divergence term for smooth tube

When weight is assigned to smooth tubes, there should always be a divergence term, for the following reason.

Let $x$ be a diffusion process that follows the stochastic differential equation

$$dx_t = f(x_t)dt + dw_t, \tag{B9}$$

where $w$ is a Wiener process. To investigate paths near a smooth curve $\phi$, let us consider the following stochastic process $x_t - \phi(t)$ (Ikeda and Watanabe, 1981; Zeitouni, 1989):

$$d(x_t - \phi(t)) = (f(x_t - \phi(t) + \phi(t)) - \dot\phi(t))dt + dw_t. \tag{B10}$$

This means that if a drift $f$ is applied to the Wiener process, and the reference frame is shifted by $\phi$, the process $x_t - \phi(t)$ which has the drift $f(\cdot + \phi) - \dot\phi$ is obtained. The weight relative to the Wiener measure can be calculated by Girsanov's formula as follows.

$$
\begin{aligned}
I_\epsilon(\phi) &\equiv \frac{P(\|x - \phi\|_T < \epsilon)}{P(\|w\|_T < \epsilon)} \\
&= \mathbb{E}\left[\exp\left(\int_0^T \left(f(w_t + \phi(t)) - \dot\phi(t)\right) \cdot dw_t - \frac{1}{2}\int_0^T \left|f(w_t + \phi(t)) - \dot\phi(t)\right|^2 dt\right)\bigg|\|w\|_T < \epsilon\right],
\end{aligned} \tag{B11}
$$

where the expectation is taken with respect to the Wiener process $w$ conditioned to $\|w\|_T \equiv \sup_{0<t<T}|w_t| < \epsilon$. We are going to evaluate the terms containing $w_t$ in the exponent on the RHS of Eq. (B11).

1. If we assume $\phi$ is a twice continuously differentiable function, then by applying Ito's product rule to $\dot\phi(t)w_t$, and using $(\forall t) |w_t| < \epsilon$,

$$\left|\int_0^T \dot\phi(t)dw_t\right| = \left|\dot\phi(T)w_T - \int_0^T w_t\ddot\phi(t)dt\right| \le A_1\epsilon, \tag{B12}$$

where $A_1$ is a positive constant independent of $\epsilon$.

2. If we assume $f$ is a twice continuously differentiable function, then by using $(\forall t) |w_t| < \epsilon$,

$$\left|\int_0^T f(w_t + \phi(t))\dot\phi(t)dt - \int_0^T f(\phi(t))\dot\phi(t)dt\right| \le A_2\epsilon, \tag{B13}$$

where $A_2$ is a positive constant independent of $\epsilon$.

3. In the similar manner as in 2,

$$\left|\int_0^T |f(w_t + \phi(t))|^2 dt - \int_0^T |f(\phi(t))|^2 dt\right| \le A_3\epsilon, \tag{B14}$$

where $A_3$ is a positive constant independent of $\epsilon$.

4. The evaluation of $\int_0^T f(w_t + \phi(t)) \cdot dw_t$ is as follows.

(a) By applying Taylor's expansion to $f(w_t + \phi(t))$,

$$\int_0^T f(w_t + \phi(t)) \cdot dw_t = \int_0^T f(\phi(t)) \cdot dw_t + \int_0^T (w_t \cdot \nabla) f(\phi(t)) \cdot dw_t + \int_0^T O(w^2) \cdot dw_t. \tag{B15}$$

(b) By applying Ito's product rule to $w_t f(\phi(t))$, and using $(\forall t)\ |w_t| < \epsilon$,

$$\int_0^T f(\phi(t)) \cdot dw_t = w_T f(\phi(T)) - \int_0^T \sum_{i,j} w_t^i \frac{\partial f_i}{\partial x_j}(\phi(t)) \dot{\phi}_j(t) dt = O(\epsilon). \tag{B16}$$

(c) Regarding the second term on the RHS of Eq. (B15), we see that

$$\int_0^T (w_t \cdot \nabla) f(\phi(t)) \cdot dw_t + \frac{1}{2} \int_0^T \nabla \cdot f(\phi(t)) dt$$

$$= \int_0^T \sum_{i,j} \frac{\partial f_i}{\partial x_j}(\phi(t)) w_t^j dw_t^i + \frac{1}{2} \int_0^T \sum_{i,j} \delta_{ij} \frac{\partial f_i}{\partial x_j}(\phi(t)) dt$$

$$= \int_0^T \sum_{i,j} \frac{\partial f_i}{\partial x_j}(\phi(t)) \left( w_t^j dw_t^i + \frac{1}{2} \delta_{ij} dt \right) = \int_0^T \sum_{i,j} \frac{\partial f_i}{\partial x_j}(\phi(t)) d\zeta_t^{ji}, \tag{B17}$$

where $\zeta_t^{ji} = \int_0^t w_s^j \circ dw_s^i$ (Stratonovich integral).

By applying Evaluations 1–4 to Eq. (B11), we obtain

$$I_\epsilon(\phi) = \exp\left( -\frac{1}{2} \int_0^T \left| f(\phi(t)) - \dot{\phi}(t) \right|^2 dt - \frac{1}{2} \int_0^T \nabla \cdot f(\phi(t)) dt \right)$$

$$\times \mathbb{E}\left[ \exp\left( O(\epsilon) + O(\epsilon^2) + \int_0^T \sum_{i,j} \frac{\partial f_j}{\partial x_i}(\phi(t)) d\zeta_t^{ji} + \int_0^T O(|w|^2) \cdot dw_t \right) \middle| \|w\|_T < \epsilon \right], \tag{B18}$$

On pages 450–451 in Ikeda and Watanabe (1981), it is shown that

$$\mathbb{E}\left[ \exp\left( c \int_0^T \sum_{i,j} \frac{\partial f_j}{\partial x_i}(\phi(t)) d\zeta_t^{ji} \right) \middle| \|w\|_T < \epsilon \right] \xrightarrow{\epsilon \to 0} 1 \quad (\forall c), \tag{B19}$$

$$\mathbb{E}\left[ \exp\left( c \int_0^T O(|w|^2) \cdot dw_t \right) \middle| \|w\|_T < \epsilon \right] \xrightarrow{\epsilon \to 0} 1 \quad (\forall c), \tag{B20}$$

and it is obvious that

$$\mathbb{E}\left[ \exp\left( cO(\epsilon) + cO(\epsilon^2) \right) \middle| \|w\|_T < \epsilon \right] \xrightarrow{\epsilon \to 0} 1 \quad (\forall c). \tag{B21}$$

They also showed that if

$$\mathbb{E}\left[\exp\left(ca_j\right)|\|w\|_T < \epsilon\right] \xrightarrow{\epsilon\to 0} 1 \quad (\forall c) \tag{B22}$$

for $j = 1, 2, \cdots, J$, then

$$\mathbb{E}\left[\exp\left(\sum_{j=1}^{J} a_j\right)\bigg|\|w\|_T < \epsilon\right] \xrightarrow{\epsilon\to 0} 1. \tag{B23}$$

By applying this to Eqs. (B20), (B19), and (B21), we deduce from Eq. (B18) that

$$I_\epsilon(\phi) \xrightarrow{\epsilon\to 0} \exp\left(-\frac{1}{2}\int_0^T \left|f(\phi(t)) - \dot{\phi}(t)\right|^2 dt - \frac{1}{2}\int_0^T \nabla \cdot f(\phi(t))dt\right). \tag{B24}$$

From evaluation 4, we also have that

$$\mathbb{E}\left[\exp\left(\int_0^T f(w_t + \phi(t)) \cdot dw_t\right)\bigg|\|w\|_T < \epsilon\right] \xrightarrow{\epsilon\to 0} \exp\left[-\frac{1}{2}\int_0^T \operatorname{div} f(\phi(t))dt\right]. \tag{B25}$$

Eq. (B25) serves as an evaluation formula for the divergence term along $\phi$ via ensemble calculation if we interpret the

expectation as an ensemble average:

$$\ln\mathbb{E}\left[\exp\left(\int_0^T f(w_t + \phi(t)) \cdot dw_t\right)\bigg|\|w\|_T < \epsilon\right] \xrightarrow{\epsilon\to 0} -\frac{1}{2}\int_0^T \operatorname{div} f(\phi(t))dt. \tag{B26}$$

The ensemble can be generated by using a Wiener process limited to the small area $\|w\|_T < \epsilon$. Taking the derivative of Eq. (B26) with respect to $\phi_i(t)$, we also obtain the formula for evaluating the derivative of the divergence term along $\phi$, as follows.

$$\frac{\mathbb{E}\left[\nabla f(\phi + w) \cdot dw \exp\left(\int_0^T f(\phi + w) \cdot dw\right)\big|\|w\|_T < \epsilon\right]}{\mathbb{E}\left[\exp\left(\int_0^T f(\phi + w) \cdot dw\right)\big|\|w\|_T < \epsilon\right]} \xrightarrow{\epsilon\to 0} -\frac{1}{2}\nabla(\operatorname{div} f)dt, \tag{B27}$$

where $(\nabla f(\phi + w), dw) = \sum_j \frac{\partial f_j(\phi + w)}{\partial \phi_i}dw_j$ can be calculated using the adjoint model $\nabla f(\phi + w)$. Although these evaluation formulas (B26) and (B27) illustrate the meaning of the divergence term, they seem too expensive to be used in the 4D-Var iterations.

## Appendix C: Estimator for the divergence term

Cost functions in Eqs. (42) and (41) utilise the derivative of the drift term $f(x)$, and thus the gradient of the term contains

the second derivative of $f(x)$, whose algebraic form is difficult to obtain in high-dimensional systems. Here, we propose an alternative form using Hutchinson's trace estimator (Hutchinson, 1990), which approximates the trace of matrix $\mathbb{E}[\xi^T A\xi] = \operatorname{tr}(A)$ using a stochastic vector whose components are independent, identically distributed stochastic variables that take value $\pm 1$ with probability 0.5.

A realisation of the cost function is given as

$$\hat{J}_{\text{tube}}(\phi|y) = \frac{1}{2\sigma_b^2}|\phi_0 - x_b|^2 + \sum_{m \in M} \frac{1}{2\sigma_o^2}|\phi_m - y_m|^2$$

$$+ \delta_t \sum_{n=1}^{N} \left( \frac{1}{2\sigma^2}\left|\frac{\phi_n - \phi_{n-1}}{\delta_t} - f(\phi_{n-1})\right|^2 + \frac{1}{2}\xi_{n-1}^T b^{-1}\left[f(\phi_{n-1} + b\xi_{n-1}) - f(\phi_{n-1})\right]\right), \tag{C1}$$

where $b$ is a small number. Notice that $\hat{J}_{\text{tube}}(\phi|y)$ is a stochastic variable that satisfies

$$\mathbb{E}\left[\hat{J}_{\text{tube}}(\phi|y)\right] = J_{\text{tube}}(\phi|y). \tag{C2}$$

If the adjoint of $f$ is at hand, the gradient of the stochastic cost function is given as

$$\nabla_{\phi_n}\hat{J}_{\text{tube}}(\phi|y) = \frac{1}{\sigma_b^2}(\phi_0 - x_b)\delta_{0,n} + \sum_{m \in M}\frac{1}{\sigma_o^2}(\phi_m - y_m)\delta_{m,n}$$

$$+ \frac{1}{\sigma^2}\left(\frac{\phi_n - \phi_{n-1}}{\delta_t} - f(\phi_{n-1})\right) \qquad (n > 0)$$

$$+ \frac{\delta_t}{\sigma^2}\left(-\frac{1}{\delta_t} - \left(\frac{\partial f}{\partial \phi_n}(\phi_n)\right)^T\right)\left(\frac{\phi_{n+1} - \phi_n}{\delta_t} - f(\phi_n)\right) \qquad (n < N)$$

$$+ \frac{\delta_t}{2}\left[\left(\frac{\partial f}{\partial \phi_n}(\phi_n + b\xi_n)\right)^T b^{-1}\xi_n - \left(\frac{\partial f}{\partial \phi_n}(\phi_n)\right)^T b^{-1}\xi_n\right]. \qquad (n < N) \tag{C3}$$

The iterations similar to Eq. (48), $\phi^{(k+1)} = \phi^{(k)} - \alpha\nabla\hat{J}_{\text{tube}}$, will work.

*Competing interests.* The authors declare that they have no competing interests.

*Acknowledgements.* The author is grateful to the referees for their comments which helped improve the readability of the paper. This work was partly supported by MEXT KAKENHI Grant-in-Aid for Scientific Research on Innovative Areas JP15H05819. All the numerical simulations were performed on the JAMSTEC SC supercomputer system.

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
