# Peer review of "The Onsager-Machlup functional for data assimilation"

_Nonlinear Processes in Geophysics, 2017_

## Referee Comment (RC1) · Anonymous Referee #1 · 7 Sep 2017

Review of npg-2017-44 "The Onsager–Machlup functional for data assimilation"

**General remark**
For an ordinary reader who is not already familiar with the Onsager–Machlup (OM) functional, Girsanov formula and Radon–Nikodym derivative, the formulations and related derivations presented in this paper are too sketch to follow. My specific comments are given below.

**Specific Comments**
1. The first sentence in the abstract is confusing or inaccurate, because one does not have to resort to the more advanced and more difficult OM functional, as long as the stochastic differential equation (SDE) is to be solved in a time-discretized form (rather than time-continuous form) for data assimilation application. Nonlinear SDEs can rarely be solved analytically in time-continuous form. Although the OM functional is useful and important for rigorous theoretical considerations when the time-continuous limit is applied to a time-discretized form of quadratic cost function, the time-continuous limit can be derived formally (or intuitively) without considering the OM functional, as shown in (5.36)-(5.40) on pages 155-156 of Jazwinski (1970: *Stochastic Processes and Filtering Theory*). Since the SDEs are actually solved in time-continuous forms in this paper (as well as in most data assimilation studies), the importance and utility of the OM functional for real data assimilation appear to be overstated in this paper.

2. The noise intensity $\sigma$ in (1) is a constant rather than a function of $x_t$. In this case, as explained at the end of section 4.6 (pages 119-120) of Jazwinski (1970), the associated Ito integral and Stratonovich integral are identical. Thus, as a stochastic differential equation, (1) can be viewed either as an Ito equation or its equivalent Stratonovich equation. The authors may need to clarify this point.

3. Eq. (13) is derived from (A2) by the scaling for a smooth curve, but the scaling $x_n - x_{n-1} = O(\delta_t)$ is not explained in Appendix A.

4. It appears that (21) is derived from (13) and (19) with $\phi$ changed into $x$, but it is not clear why $\phi$ can be changed into $x$.

5. It is not shown and unclear how (22) is derived.

6. Due to the questions in above comments 3-4, it is not clear whether the four schemes considered in section 2 all converge to the same time-continuous limit. If the answer is yes, then the differences between the numerical results obtained from the four schemes for each example in section 3 are caused the differences in discretization, and these differences should diminish as $\delta_t$ approaches to zero. To verify this numerically, the authors need to show for each example that the differences between the numerical results obtained from the four schemes become increasingly small as $\delta_t$ decreases toward zero.

7. Section 6.3.2 of Law et al. (2015) is cited for the derivation of divergence term in Appendix B. I checked but found that there are only 5 chapters in Law et al. (2015).

8. The formulation on the line above (B7) appears to be for $\delta\mu_o/\delta x$ or $d\mu_o/dx$ [that is, the

variation of $\mu_0$ with respect to variation of $x(t)$ for $0 \le t \le T$] rather than for the Wiener measure $\mu_0$ itself. Similarly, $\mu$ should be $\delta\mu/\delta x$ or $d\mu/dx$ on the left-hand side of (B7) and (B8). Correct?

9. As a reader, I would like to see the detailed step-by-step derivations (with adequate interpretations) of (B11)-(B14).

---

## Author Comment (AC1) · 13 Sep 2017

The author really appreciates the 1st reviewer for careful reading of the manuscript including the Appendix. Although the revision according to the reviewer's general remark will take some time, I provide quick and tentative replies to the specific comments in order to enhance the discussion.

1. *The first sentence in the abstract is confusing or inaccurate, because one does not have to resort to the more advanced and more difficult OM functional, as long as the stochastic differential equation (SDE) is to be solved in a time-discretized form (rather than time-continuous form) for data assimilation application. Nonlinear SDEs can rarely be solved analytically in time-continuous form. Although the OM functional is*

*useful and important for rigorous theoretical considerations when the time-continuous limit is applied to a time-discretized form of quadratic cost function, the time-continuous limit can be derived formally (or intuitively) without considering the OM functional, as shown in (5.36)-(5.40) on pages 155-156 of Jazwinski (1970: Stochastic Processes and Filtering Theory). Since the SDEs are actually solved in time-continuous forms in this paper (as well as in most data assimilation studies), the importance and utility of the OM functional for real data assimilation appear to be overstated in this paper.*

All of the SDEs are actually solved in time-discrete form in this paper, and thus I am sure that the importance of OM functional in data assimilation is properly illustrated. The derivation in Jazwinski (1970) is valid for the assignment of each path probability, but we should be careful when we consider an optimization problem. If we solve the optimization problem with Jazwinski's strategy, that should lead to curves like "E" or "T" in Fig.3 and Fig.5, which is clearly less meaningful than "ED" or "TD".

Even in a discrete-time setting, a path drawn with model error term is generally not differentiable in time direction since the random term on each time slice adds an independent noise, and thus a smoother, whose object is smooth functions, cannot optimize the paths itself. What we can do is to draw smooth curves and compare the densities of paths in their $\epsilon$-neighborhoods, which is shown in the manuscript.

*2. The noise intensity $\sigma$ in (1) is a constant rather than a function of $x_t$ . In this case, as explained at the end of section 4.6 (pages 119-120) of Jazwinski (1970), the associated Ito integral and Stratonovich integral are identical. Thus, as a stochastic differential equation, (1) can be viewed either as an Ito equation or its equivalent Stratonovich equation. The authors may need to clarify this point.*

I agree with the reviewer's comment that there is no need to distinguish the Ito integral from the Stratonovich integral with regard to the discretization of the stochastic differential equation (SDE). Note that the distinction is applied in the manuscript only to the discretization of the OM functional, not to the SDE itself, because the quadratic term

in the former contains the product of the noise $dw_t$ and the process dependent term $f(x_t)$.

3. *Eq. (13) is derived from (A2) by the scaling for a smooth curve, but the scaling $x_n - x_{n-1} = O(\delta_t)$ is not explained in Appendix A.*

In the case of a smooth curve, there is no stochastic term, and thus $x_n - x_{n-1}$ is the product of a bounded function $f(x_{n-1})$ and $\delta_t$, which results in a value with $O(\delta_t)$. I will mention that in the revised manuscript.

4. *It appears that (21) is derived from (13) and (19) with $\phi$ changed into $x$, but it is not clear why $\phi$ can be changed into $x$.*

The symbol $x$ in (21) represents either $\phi$ or $x$. Since the notation was confusing, I will change the symbol $x$ to $\chi$ (or something) in (21) and the related expressions.

5. *It is not shown and unclear how (22) is derived.*

The four cases are the conceivable combinations of the timing of the drift term $f(x_t)$ and the presence or absence of the divergence term. Equation (22) is just one of them. I will mention that in the revised manuscript.

6. *Due to the questions in above comments 3-4, it is not clear whether the four schemes considered in section 2 all converge to the same time-continuous limit. If the answer is yes, then the differences between the numerical results obtained from the four schemes for each example in section 3 are caused the differences in discretization, and these differences should diminish as $\delta_t$ approaches to zero. To verify this numerically, the authors need to show for each example that the differences between the numerical results obtained from the four schemes become increasingly small as $\delta_t$ decreases toward zero.*

Not all of them converge to the same limit. Rather, the schemes that I judged to be applicable (see table 1) are expected to converge to the common answers, which you can see clearly in Figs. 2 to 5. I will mention that in the revised manuscript.

7. *Section 6.3.2 of Law et al. (2015) is cited for the derivation of divergence term in Appendix B. I checked but found that there are only 5 chapters in Law et al. (2015).*

I am afraid you refer to the preprint version of Law et al. (2015) in arxiv. Please refer to the commercially published version, which has 9 chapters.

8. *The formulation on the line above (B7) appears to be for $\delta\mu_0/\delta x$ or $d\mu_0/dx$ [that is, the variation of $\mu_0$ with respect to variation of $x(t)$ for $0 \leq t \leq T$] rather than for the Wiener measure $\mu_0$ itself. Similarly, $\mu$ should be $\delta\mu/\delta x$ or $d\mu/dx$ on the left-hand side of (B7) and (B8). Correct?*

As the reviewer pointed out, the expression for the measure was inaccurate. I will change $\mu_0$ to $\mu_0(d\Omega)$, $\mu$ to $\mu(d\Omega)$ as well, and add the explanation as follows.

We divide a time interval $[0,T]$ to $N$ equal segments $[(n-1)\delta_t, n\delta_t]$, $n = 1, 2, \cdots, N$ with $T = N\delta_t$, and consider a cylinder set $\Omega \equiv \Omega_1 \times \Omega_2 \times \cdots \times \Omega_N$ where $\Omega_n \subset \mathbb{R}^D$ is on time slice $t = n\delta_t$. We can define a measure for the cylinder set $\Omega$ as

$$\mu_0(\Omega) = \int_{\Omega_1} p(\delta_t, x_1 - x_0)dx_1 \int_{\Omega_2} p(\delta_t, x_2 - x_1)dx_2 \cdots \int_{\Omega_N} p(\delta_t, x_N - x_{N-1})dx_N,$$

$$p(\delta_t, x' - x) = \frac{1}{\sqrt{2\pi\delta_t}} e^{-\frac{|x'-x|^2}{2\delta_t}}.$$

The Wiener measure is the extension of the above $\mu_0$ to an infinite number of segments ($\delta_t \to 0$). We write it symbolically as

$$\mu_0(\Omega) = \int_{\Omega} \mu_0(d\Omega),$$

$$\mu_0(d\Omega) \propto \exp\left[-\frac{1}{2}\int_0^T \left|\frac{dx}{dt}\right|^2 dt\right] d\Omega.$$

9. *As a reader, I would like to see the detailed step-by-step derivations (with adequate interpretations) of (B11)-(B14).*

Thank you for the interest. I will append explanations to the derivation.

---

## Referee Comment (RC2) · Anonymous Referee #2 · 14 Sep 2017

I enjoyed reading this paper. I found it to be well written, well structured and presented in such a way that the results can be well understood and re-created.

In addition to the suggestions from the other referee I would suggest:

1. I find the abstract to be a bit vague and think you may want to state more clearly what is done in the paper. For example mention that you implement 4DVAR. The way it's written makes it unclear if you're referring to things covered in the paper or in the literature.

2. You mention later on that the methods involving the divergence of f would be difficult to implement for large dimension and 4DVAR due to the need to compute the derivative of div f. I think it would be interesting if you could mention this when you introduce your

models, or show the gradient of the cost function for 4DVAR and highlight the need of this extra term. E.g. after Eq 25 you could have a "where delJ_tube = ... (26)"

3. P5L23 "You can see" -> "It is clear"

4. Yaxis labels in Fig 1 are missing and a number at the origin is not clear.
* * *

---

## Author Comment (AC2) · 5 Oct 2017

The author sincerely appreciates the 1st referee's careful review of the Manuscript and Appendix. The author's responses to the referee's comments are as follows:

**General remark**   *For an ordinary reader who is not already familiar with the Onsager–Machlup (OM) functional, Girsanov formula and Radon–Nikodym derivative, the formulations and related derivations presented in this paper are too sketch to follow.*

Considering the reviewer's general remark, a self-contained explanation of the paper's fundamental concepts is provided as follows. Probably the most difficult and important part of this paper is establishing why the divergence term is needed in the cost function for 4D-Var. Thus, I have appended a derivation to the Introduction without explicitly using the above concepts (the OM functional, Girsanov formula, or Radon–Nikodym derivative). (see introduction 1.2 in the revised ms, starting from Page 3 Lines 20)

I have also added an explanation on the path probability. (see introduction 1.1 in the revised ms, starting from Page 2 Line 17)

The corresponding mathematical concepts are as follows. If you apply a drift $f$ to a random walk, and shift the reference frame by $\phi$ as in Eq. (15) in the revised ms, then you get a weight relative to a random walk $y$, which is a Radon–Nikodym derivative, as in Eq (16). This is nothing but an application of Girsanov formula (e.g. Example 8.6.9 in Øksendal (2003)). The exponent in the probability density of $\phi$ in Eq. (26) is called the Onsager–Machlup functional. I believe these explanations will help readers understand the basic concept.

In accordance with these changes, the beginning of the introduction has been simplified. (see introduction in the revised ms, starting from Page 1 Line 11)

**1.**   *The first sentence in the abstract is confusing or inaccurate, because one does not have to resort to the more advanced and more difficult OM functional, as long as the stochastic differential equation (SDE) is to be solved in a time-discretized form (rather than time-continuous form) for data assimilation application. Nonlinear SDEs can rarely be solved analytically in time-continuous form. Although the OM functional is useful and important for rigorous theoretical considerations when the time-continuous limit is applied to a time-discretized form of quadratic cost function, the time-continuous limit can be derived formally (or intuitively) without considering the OM functional, as shown in (5.36)-(5.40) on pages 155-156 of Jazwinski (1970: Stochastic Processes and Filtering Theory). Since the SDEs are actually solved in time-continuous forms in this paper (as well as in most data assimilation studies), the importance and utility of the OM functional for real data assimilation appear to be overstated in this paper.*

All of the SDEs are actually solved in **time-discrete form** in this study; thus, I am sure that the importance of the OM functional in data assimilation is properly illustrated. The derivation in Jazwinski (1970) is valid for the assignment of each path probability, but we should be careful when we consider an optimisation problem. Solving the optimisation problem with Jazwinski's strategy should lead to curves like 'E' or 'T' in Fig. 3 and Fig. 5, which are clearly less meaningful than 'ED' or 'TD'.

Even in a discrete-time setting, a path drawn with a model error term is generally not differentiable in the time direction, because the random term on each time slice adds an independent noise; thus, a smoother, whose object is smooth functions, cannot optimise the paths itself. What we can do is to

draw smooth curves and compare the densities of paths in their $\epsilon$-neighbourhoods; this is shown in the manuscript.

Please also refer to the counterexample at the beginning of section 1.2 in the revised ms. (Page 4 Lines 1–10)

**2.** *The noise intensity $\sigma$ in (1) is a constant rather than a function of $x_t$ . In this case, as explained at the end of section 4.6 (pages 119-120) of Jazwinski (1970), the associated Ito integral and Stratonovich integral are identical. Thus, as a stochastic differential equation, (1) can be viewed either as an Ito equation or its equivalent Stratonovich equation. The authors may need to clarify this point.*

I agree with the reviewer's comment that there is no need to distinguish the Ito integral from the Stratonovich integral with regard to the discretisation of the stochastic differential equation (SDE). Note that in the manuscript, the distinction is applied only to the discretisation of the OM functional, not to the SDE itself, because the quadratic term in the former contains the product of the noisy term $dx_t$ and the process dependent term $f(x_t)$.

**3.** *Eq. (13) is derived from (A2) by the scaling for a smooth curve, but the scaling $x_n - x_{n-1} = O(\delta_t)$ is not explained in Appendix A.*

According to the reviewer's suggestion, I have added the following explanation.

'In the case of a smooth curve, there is no stochastic term, and thus $\psi_n - \psi_{n-1}$ is the product of a bounded function $f(\psi_{n-1})$ and $\delta_t$, which results in a value with $O(\delta_t)$'. (Page 15 Lines 12–13)

**4.** *It appears that (21) is derived from (13) and (19) with $\phi$ changed into $x$, but it is not clear why $\phi$ can be changed into $x$.*

The symbol $x$ in (21) represents either $\phi$ or $x$. Because the notation was confusing, I have changed the symbol $x$ to $\psi$ in (21) and the related expressions. (Page 8 Lines 1–12)

**5.** *It is not shown and unclear how (22) is derived.*

The four cases are the conceivable combinations of the timing of the drift term $f(x_t)$ and the presence or absence of the divergence term. Equation (22) is just one of them. I have added the following sentence.

'As a proof-of-concept described in these sections, we will test all the cases with conceivable combinations of the timing of the drift term $f(x_t)$ and the presence or absence of the divergence term'. (Page 7 Lines 21–22)

**6.** *Due to the questions in above comments 3-4, it is not clear whether the four schemes considered in section 2 all converge to the same time-continuous limit. If the answer is yes, then the differences between the numerical results obtained from the four schemes for each example in section 3 are caused the differences in discretization, and these differences should diminish as $\delta_t$ approaches*

*to zero. To verify this numerically, the authors need to show for each example that the differences between the numerical results obtained from the four schemes become increasingly small as $\delta_t$ decreases toward zero.*

Not all of them converge to the same limit. Rather, the schemes that I judged to be applicable (see table 1) are expected to converge to the common answers, which you can see clearly in Figs. 2 through 5. I have mentioned it in the conclusion.

'Table 1 lists the discretisation schemes which were found to be applicable, i.e. those expected to converge to the same result as the reference solution'. (Page 14 Lines 9–10)

**7.** *Section 6.3.2 of Law et al. (2015) is cited for the derivation of divergence term in Appendix B. I checked but found that there are only 5 chapters in Law et al. (2015).*

It appears that you are referring to the preprint version of Law et al. (2015) in arXiv. Please refer to the commercially published version, which has nine chapters.

**8.** *The formulation on the line above (B7) appears to be for $\delta\mu_0/\delta x$ or $d\mu_0/dx$ [that is, the variation of $\mu_0$ with respect to variation of $x(t)$ for $0 \leq t \leq T$] rather than for the Wiener measure $\mu_0$ itself. Similarly, $\mu$ should be $\delta\mu/\delta x$ or $d\mu/dx$ on the left-hand side of (B7) and (B8). Correct?*

As the reviewer pointed out, the expression for the measure was inaccurate. I have changed $\mu_0$ to $\mu_0(dx) = \cdots dx$, and $\mu$ to $\mu(dx) = \cdots dx$. (Page 16 Lines 7–11)

**9.** *As a reader, I would like to see the detailed step-by-step derivations (with adequate interpretations) of (B11)-(B14).*

Thank you for the interest. I have appended detailed explanations to Appendix B2. (Page 16 Line 13 to Page 19 Line 6)

**References**

Øksendal, B.: Stochastic Differential Equations, Springer-Verlag Berlin Heidelberg, 6th edn., 2003.

[revised manuscript text omitted]
}_y \left[ e^{-J(\phi,y)} \middle| (\forall n) \; |y_n| < \epsilon \right], \tag{17}$$

$$J(\phi,y) \equiv -\frac{\delta_t}{2} \sum_{n=1}^{N} \left[ \left( \frac{\phi_n - \phi_{n-1}}{\delta_t} - f(y_{n-1} + \phi_{n-1}) \right)^2 + 2 \left( \frac{\phi_n - \phi_{n-1}}{\delta_t} - f(y_{n-1} + \phi_{n-1}) \right) \left( \frac{y_n - y_{n-1}}{\delta_t} \right) \right] \tag{18}$$

where $\mathbb{E}_y$ denotes the ensemble averaging of the random walks denoted by $y$, each of which follows the route $(y_0, y_1, \cdots, y_N)$, and satisfies $|y_n| < \epsilon$ for all $n$.

Because $y_{n-1}$ is small, we can apply the expansion:

$$f(y_{n-1} + \phi_{n-1}) = f(\phi_{n-1}) + f'(\phi_{n-1})y_{n-1} + O(y^2), \tag{19}$$

where $f'$ is the derivative of $f$. Let us accept that the following average containing the higher order terms $O(y^2)$ converges (see Eq. (B20)).

$$\mathbb{E}_y \left[ e^{\sum_{n=1}^{N} O(y^2)(y_n - y_{n-1})} \middle| (\forall n) \; |y_
[revised manuscript text omitted]

---

## Author Comment (AC3) · 5 Oct 2017

The author sincerely appreciates the 2nd referee's careful reading of the manuscript; the encouraging suggestions are also appreciated. The author's responses to the referee's comments are as follows:

**1.** *I find the abstract to be a bit vague and think you may want to state more clearly what is done in the paper. For example mention that you implement 4DVAR. The way its written makes it unclear if youre referring to things covered in the paper or in the literature.*

As the reviewer pointed out, the abstract did not properly represent what the main text described. I have modified the abstract to emphasise 4D-Var and the divergence term. (see abstract in the revised ms)

**2.** *You mention later on that the methods involving the divergence of f would be difficult to implement for large dimension and 4DVAR due to the need to compute the derivative of div f. I think it would be interesting if you could mention this when you introduce your models, or show the gradient of the cost function for 4DVAR and highlight the need of this extra term. E.g. after Eq 25 you could have a "where delJ_tube = ... (26)"*

According to the reviewer's suggestion, a note on the gradient has been added. (Page 9 Lines 1–7)

**3.** *P5L23 'You can see' → 'It is clear'*

Done. (Page 10 Line 7)

**4.** *Yaxis labels in Fig 1 are missing and a number at the origin is not clear.*

Done. (see Fig. 1 in the revised ms)

[revised manuscript text omitted]
}_y \left[ e^{-J(\phi, y)} \middle| (\forall n) \; |y_n| < \epsilon \right], \tag{17}$$

$$J(\phi, y) \equiv -\frac{\delta_t}{2} \sum_{n=1}^{N} \left[ \left( \frac{\phi_n - \phi_{n-1}}{\delta_t} - f(y_{n-1} + \phi_{n-1}) \right)^2 + 2 \left( \frac{\phi_n - \phi_{n-1}}{\delta_t} - f(y_{n-1} + \phi_{n-1}) \right) \left( \frac{y_n - y_{n-1}}{\delta_t} \right) \right] \tag{18}$$

where $\mathbb{E}_y$ denotes the ensemble averaging of the random walks denoted by $y$, each of which follows the route $(y_0, y_1, \cdots, y_N)$, and satisfies $|y_n| < \epsilon$ for all $n$.

Because $y_{n-1}$ is small, we can apply the expansion:

$$f(y_{n-1} + \phi_{n-1}) = f(\phi_{n-1}) + f'(\phi_{n-1}) y_{n-1} + O(y^2), \tag{19}$$

where $f'$ is the derivative of $f$. Let us accept that the following average containing the higher order terms $O(y^2)$ converges (see Eq. (B20)).

$$\mathbb{E}_y \left[ e^{\sum_{n=1}^{N} O(y^2)(y_n - y_{n-1})} \middle| (\forall n) \; |y_
[revised manuscript text omitted]

---

## Author Response (AR2)

The author sincerely appreciates the editor's careful review of Manuscript and Responses. The author's responses to the editor's comments are as follows:

**1)** *With respect to Comment #2 from Rev. 1, you mention in your written response that you agree with a point raised by Rev 1 here but then do not point out if and how you may have revised the manuscript in response? Can you please clarify? This information may also help the second round of review.*

As the editor pointed out, I had not revised the manuscript on this point in the earlier draft. I have revised the manuscript by adding the following sentence.

'Note that there is no need to distinguish the Ito integral from the Stratonovich integral with regard to the discretisation of the SDE, because the noise intensity is a constant'. (Page 2 Lines 8–9)

**2)** *As to Comment #1 from Rev. 2, thank you for making the Abstract more informative and explicit. Yet I feel there is more you could do in the Abstract, see for example the 2nd sentence, to clarify "if you are referring to things covered in the paper or in the literature".*

Following the suggestion by the editor, I have revised the abstract to clarify what I have done in this study as follows.

[revised manuscript text omitted]